# Dietary inulin alters the gut microbiome, enhances systemic metabolism and reduces neuroinflammation in an *APOE4* mouse model

Jared D. Hoffman[1,2], Lucille M. Yanckello[1,2], George Chlipala[3], Tyler C. Hammond[1,4], Scott D. McCulloch[5], Ishita Parikh[1,6], Sydney Sun[1], Josh M. Morganti[1,4,7], Stefan J. Green[3], Ai-Ling Lin[1,2,4,8]*

**1** Sanders-Brown Center on Aging, University of Kentucky, Lexington, Kentucky, United States of America, **2** Department of Pharmacology and Nutritional Science, University of Kentucky, Lexington, Kentucky, United States of America, **3** Research Resources Center, University of Illinois at Chicago, Chicago, Illinois, United States of America, **4** Department of Neuroscience, University of Kentucky, Lexington, Kentucky, United States of America, **5** Metabolon Inc., Durham, North Carolina, United States of America, **6** Oklahoma Medical Research Foundation, Oklahoma City, Oklahoma, United States of America, **7** Spinal Cord and Brain Injury Research Center, University of Kentucky, Lexington, Kentucky, United States of America, **8** F. Joseph Halcomb III, M.D. Department of Biomedical Engineering, University of Kentucky, Lexington, Kentucky, United States of America

* ailing.lin@uky.edu

**Data Availability Statement:** The microbiome gene amplicon sequence data are available atNCBI

## Abstract

The apolipoprotein ε4 allele (*APOE4*) is the strongest genetic risk factor for Alzheimer's disease (AD). *APOE4* carriers develop systemic metabolic dysfunction decades before showing AD symptoms. Accumulating evidence shows that the metabolic dysfunction accelerates AD development, including exacerbated amyloid-beta (Aβ) retention, neuroinflammation and cognitive decline. Therefore, preserving metabolic function early on may be critical to reducing the risk for AD. Here, we show that inulin increases beneficial microbiota and decreases harmful microbiota in the feces of young, asymptomatic *APOE4* transgenic (E4FAD) mice and enhances metabolism in the cecum, periphery and brain, as demonstrated by increases in the levels of SCFAs, tryptophan-derived metabolites, bile acids, glycolytic metabolites and scyllo-inositol. We show that inulin also reduces inflammatory gene expression in the hippocampus. This knowledge can be utilized to design early precision nutrition intervention strategies that use a prebiotic diet to enhance systemic metabolism and may be useful for reducing AD risk in asymptomatic *APOE4* carriers.

## Introduction

Alzheimer's disease (AD) is most common form of dementia with hallmarks of amyloid beta (Aβ) plaques and neurofibrillary tau tangles [1]. Research on AD has long been dominated by communications regarding the Aβ hypothesis and targeting Aβ accumulation in the brain through pharmacological therapies. Unfortunately, this research strategy has failed to produce any FDA-approved disease-modifying therapeutics for Alzheimer's disease despite over 900

BioProject database (PRJNA540508). All other relevant data are within the paper.

**Funding:** This research was supported by NIH grants R01AG054459 (funded by NIA and ODS), R01AG062480, NIH/CTSA grant UL1TR000117, and American Federation for Aging Research Grant #A12474 to A-LL and NIH/NIA R21AG058006 to JMM, and NIH/NIDDK Training Grant T32DK007778 to JDH and LMY. The 7T ClinScan small animal MRI scanner of the UK was funded by the S10 NIH Shared Instrumentation Program Grant (1S10RR029541-01). The content is solely the responsibility of the authors and does not necessarily represent the official views of the National Institute on Aging or the National Institutes of Health. SDM is employed by Metabolon Inc.. Metabolon Inc. provided support in the form of salary for author SDM but did not have any additional role in the study design, data collection and analysis, decision to publish, or preparation of the manuscript.

**Competing interests:** Scott McCulloch is employed by Metabolon Inc. This does not alter our adherence to PLOS ONE policies on sharing data and materials.

interventional studies completed and registered in the ClinicalTrials.gov database [2]. Accumulating evidence suggests that systemic metabolic dysfunction, oxidative stress and inflammation contribute greatly to AD risk as individuals with insulin resistance, hyperlipidemia, obesity, type 2 diabetes, and other metabolic diseases are at an increased risk for AD [3]. The metabolic deficits and increased inflammation and oxidative stress may occur decades before Aβ and tau retention become evident [4, 5].

The metabolism-induced AD acceleration is specifically true for carriers of the *ε4* allelic variant of the *APOE* gene (HGNC:613), the strongest genetic risk factor for late-onset AD [6]. Decades before the aggregation of Aβ, asymptomatic *APOE4* carriers already show metabolic deficits that may lead to disease progression. Young and middle-aged cognitively normal *APOE4* carriers have reduced glucose uptake in the brain [7]. Similarly, *APOE4* transgenic mice have mitochondrial dysfunction, insulin-signaling impairment, and alterations of the pentose phosphate pathway (PPP) [8, 9]. Systemically, *APOE4* carriers have significantly elevated fasting glucose and insulin levels along with an increased risk of metabolic syndrome [10] and chronic low-grade inflammation [11]. Accordingly, it is critical for *APOE4* carriers to reduce their risk of AD by systemically preserving metabolic function and reducing inflammation.

Emerging evidence shows that the gut microbiome plays a critical role in modulating metabolism, immune function, and Aβ deposition in the host [12–15], implicating it in the modulation of AD pathology [16]. The three most studied categories of metabolites produced in host-microbiota interactions are short-chain fatty acids (SCFAs), bile acids, and tryptophan-derived metabolites [17]. SCFAs, including butyrate, acetate and propionate, have a dramatic impact on metabolic function [18]. Bile acids significantly impact pathways involved in host cholesterol, lipid and glucose metabolism, and inflammation, and have the potential to alter the host immunity [19] and circadian rhythms [20]. Indole-3-propionic acid, a tryptophan-derived metabolite, can inhibit Aβ fibril formation in neurons and neuroblastoma cells [21]. On the other hand, Aβ and lipopolysaccharides can be secreted by some gut microbiota, activating microglia and leading to neuroinflammation [22].

In this study, our goal was to determine whether controlling gut microbial composition and activity via a dietary intervention can protect systemic metabolic functions through the gut-brain axis in asymptomatic *APOE4* mice compared to their *APOE3* littermates. We used a dietary supplement containing Inulin, a non-digestible carbohydrate fiber fermented in the gastrointestinal tract. Inulin is a well-studied prebiotic compound consisting of indigestible fiber that stimulates the growth and activity of SCFA-producing bacteria [23]. In addition, inulin has been demonstrated to increase glucose sensitivity, decrease blood cholesterol and oxidative stress, and prevent neurodegeneration [14, 15, 24]. We hypothesized that inulin would alter the gut microbiome, enhance systemic metabolism and reduce neuroinflammation in the asymptomatic *APOE4* mice.

## Methods

### Experimental design

We used a C57BL/6 mouse model which accumulates human Aβ$_{42}$ due to coexpression of 5 familial-AD (5xFAD) mutations in conjunction with human targeted replacement *APOE* (*ε4* in the *E4*FAD line and *ε3* in the *E3*FAD line) which typically do not develop AD symptoms until 8–10 months of age [25]. We obtained the breeders from Dr. Mary Jo LaDu of the University of Illinois at Chicago [26], and established our own colony at the University of Kentucky. Each mouse was genotyped to verify their *APOE* and FAD genotype via Transnetyx Inc. (Cordova, TN, USA) after weaning. The experimental diet treatments began when mice were

three months of age, before they started to show Aβ retention or cognitive impairment. We fed *E4*FAD mice either a prebiotic diet containing inulin that is fermentable (*E4*FAD-Inulin) or a control diet containing cellulose that is nonfermentable (*E4*FAD-Control) for a period of 16 weeks [27]. As an additional control, we fed *E3*FAD mice the control diet (*E3*FAD-Control) to compare the results from the *APOE ε4* genotype with the more common *APOE ε3* genotype, which is considered risk-neutral for AD.

Table 1 shows the composition of the control and inulin diets. We fed the mice 8% inulin because it has been shown that 8% inulin increased cecal contents, produced more SCFA, and increased the amount of bacterial enzymes in the cecum compared to 4% inulin [28], which is consistent with human studies that 8% fiber (40 g of fiber per day) was considered the maximum tolerable and beneficial quantity to the western human organism [29].

The mice had ad libitum access to the diets, and we measured the weight of the remaining food biweekly to estimate the food intake during the 16-week feeding period. The bodyweight was also measured accordingly at the same time.

We determined the sample size (N = 15/group, M:F = 50:50) via power analysis to ensure comparison at a 0.05 level of significance and 90% chance of detecting a true difference of each measured variable between the three groups. Each mouse was housed individually to avoid feces exchange [30]. The mice were weighed biweekly and given *ad libitum* access to food and water. After 16-weeks of feeding (when the mice reached 7 months of age), we collected fecal samples, assessed cognitive functions and measured *in vivo* brain metabolites before sacrificing the mice. Mice were euthanized by $CO_2$ inhalation and decapitated. Cecum, blood and brain tissues were collected thereafter for metabolomics and other biochemical analyses (e.g., proinflammatory gene expression and Aβ immunohistochemical staining). All experimental procedures were performed according to NIH guidelines and approved by the Institutional Animal Care and Use Committee (IACUC) at the University of Kentucky (UK).

## Behavior assessments

We performed the radial arm water maze (RAWM) task to measure both spatial working memory and spatial reference memory following a 2-day testing paradigm [31]. A staggered training schedule was used, running the mice in cohorts of ten mice, while alternating the different cohorts through the trials over day 1 and day 2 of the test. This alternating protocol was used to avoid the learning limitations imposed by mass sequential trials and to avoid fatigue that may result from consecutive trials. Day 1 is the "learning" phase where mice went through three blocks (Blocks 1–3; 5 trials in each block) to test learning and short-term spatial memory. Day 2 is the "recall" phase where mice went through three additional blocks (Blocks 4–6) to test long-term memory after a 24-hour retention period to locate the platform. It is expected that after the two-day training, the mice with intact memory can find the platform with minimal errors. Geometric extra-maze visual cues were fixed throughout the study on three sides

**Table 1. Composition of the control and inulin diets.**

| Diet | Prebiotic Diet | Control Diet |
|---|---|---|
| Protein % | 18.2 | 18.2 |
| Carbohydrates % | 67.8 | 60.2 |
| Fat % | 7.1 | 7.1 |
| Fiber % | 8.0 (Inulin) | 8.0 (Cellulose) |
| Energy (kcal/g)* | 4.08 | 3.78 |

* Energy (kcal/g)—Sum of decimal fractions of protein, fat and carbohydrate x 4, 9, and 4 kcal/g, respectively.

of the curtains. Visual platform trials were included in the training and were used to determine if visual impairment could be a cofounding variable. Mouse performance was recorded by EthoVision XT 8.0 video tracking software (Noldus Information Technology) and data was analyzed by the EthoVision software for the number of incorrect arm entries which are defined as errors. The video was reviewed for each mouse to ensure that the mice did not employ non-spatial strategies, such as chaining, to solve the task.

## Proton magnetic resonance spectroscopy ($^1$H-MRS)

$^1$H-MRS was conducted on a 7T ClinScan MR scanner (Siemens, Germany) at the Magnetic Resonance Imaging & Spectroscopy Center of UK. MRS was utilized to measure metabolites in the hippocampus of a subset (n = 8/group) of mice. Mice were anesthetized with 4.0% iso-flurane for induction followed by a 1.5% isoflurane and oxygen mixture for maintenance using a facemask. Heart rate (80–120 bpm), respiration rate, rectal temperature (37 ± 0.5˚C), and water bath temperature (45–50˚C) for body temperature maintenance were monitored throughout the entire scan. The following metabolites were measured: alanine, total choline, glutamate-glutamine complex, myo-inositol, scyllo-inositol, lactate, NAA, phosphocreatine, total creatine, and taurine [32]. The following were used for a water-suppressed spectrum to test for these metabolites: TR = 1500 ms, TE = 135 ms, spectral width = 60 Hz, and average = 400. A voxel (2.0 mm x 5.0 mm x 1.3 mm) is placed over the bilateral hippocampus. Next, a non-water suppressed spectra is performed with 10 averages. Both of these spectra were processed using the LCModel software to find the absolute concentration of the metabolites. To quantify the concentrations of the metabolites, the following equation was utilized: $[m] = (S_m/S_{water})[water]C_nC_{av}$ where $[m]$ is the metabolite concentration, $S_m$ is the metabolite intensity acquired from MRS, $S_{water}$ is the water intensity acquired from MRS, [water] is the concentration of water (55.14mM at 310K), $C_n$ is the correction for the number of equivalent nuclei for each resonance, and $C_{av}$ is the correction for the number of averages [31].

## Gut microbiome analysis

**Fecal DNA amplification.** Fecal samples were collected from all mice (n = 15/group) and frozen at -80˚C until further use. A PowerSoil DNA Isolation Kit (MO BIO Laboratories, Inc.) was used for fecal DNA extraction, according to the manufacturer's protocol. Genomic DNA was PCR amplified with primers CS1_515F and CS2_926R targeting the V4-V5 regions of microbial 16S rRNA genes using a two-stage "targeted amplicon sequencing" protocol [33]. First stage amplifications were performed with the following thermocycling conditions: 95˚C for 3 minutes, 28 cycles of 95˚C for 45 seconds, 55˚C for 45 seconds, 72˚C for 90 seconds and final elongation at 72˚C for 10 minutes. Barcoding was performed using a second-stage PCR amplification with Access Array Barcode Library for Illumina Sequencers (Fluidigm, South San Francisco, CA; Item# 100–4876). The pooled libraries, with a 15% phiX spike-in, were loaded on a MiSeq v3 flow cell, and sequenced using an Illumina MiSeq sequencer, with paired-end 300 base reads. Fluidigm sequencing primers, targeting the CS1 and CS2 linker regions, were used to initiate sequencing. De-multiplexing of reads was performed on the instrument. Second stage PCR amplification and library pooling was performed at the University of Illinois at Chicago Sequencing Core. Sequencing was performed at the W.M. Keck Center for Comparative and Functional Genomics at the University of Illinois at Urbana-Champaign. The gene amplicon sequence data generated as part of this study have been submitted to the NCBI BioProject database (PRJNA540508).

**Microbial analysis.** Forward and reverse reads were merged using PEAR [34]. Primer sequences were identified using Smith-Watermann alignment and trimmed from the

sequence. Reads lacking either primer sequence were discarded. Sequences were then trimmed based on quality scores using a modified Mott algorithm with PHRED quality threshold of p = 0.01, and sequences shorter than 300 bases after trimming were discarded. QIIME v1.8 was used to generate OTU tables and taxonomic summaries [35]. Briefly, the resulting sequence files were merged with sample information. Operational taxonomic unit (OTU) clusters were generated in a *de novo* manner using the UCLUST algorithm with a 97% similarity threshold [36]. Chimeric sequences were identified using the USEARCH61 algorithm with the GreenGenes 13_8 reference sequences [37]. Taxonomic annotations for each OTU were using the UCLUST algorithm and GreenGenes 13_8 reference with a minimum similarity threshold of 90% [36, 37]. Taxonomic and OTU abundance data were merged into a single OTU table and summaries of absolute abundances of taxa were generated for all phyla, classes, orders, families, genera, and species present in the dataset. The taxonomic summary tables were then rarefied to a depth of 10,000 counts per sample.

Shannon and Bray-Curtis indices were calculated with default parameters in R using the vegan library [38]. The rarefied species data, taxonomic level 7, were used to calculate both indices. Plots were generated in R using the ggplot2 library [39]. Significant difference among tested groups was determined using the Kruskal-Wallis one-way analysis of variance. The group significance tests were performed on the rarefied species data, taxonomic level 6 (genus), using the group_significance.py script within the QIIME v1.8 package.

## Metabolomic profiling

To determine whether changes in the gut microbial community in inulin-fed *E4*AD mice correlated with changes in systemic metabolism and microbial metabolites, we examined metabolic changes in the gut, periphery, and brain of experimental mice using metabolomics profiling. Metabolomics were performed by Metabolon Inc. (Durham, NC). Metabolon's standard solvent extraction method was used to prepare the samples for analysis via liquid chromatography/mass spectrometry (LC/MS) using their standard protocol [40].

**SCFA measurement.** For SCFA analysis in the whole blood and cecal contents, eight SCFAs were analyzed by LC-MS/MS. These were as follows: acetic acid (C2), propionic acid (C3), isobutyric acid (C4), 2-methylbutyric acid (C5), isovaleric acid (C5), valeric acid (C5), and caproic acid (C6). Both sets of samples are stable labelled with internal standards and homogenized in an organic solvent. The samples are then centrifuged followed by an aliquot of the supernatant used to derivatize to form SCFA hydrazides. This reaction mixture is subsequently diluted, and an aliquot is injected into an Agilent 1290/AB Sciex QTrap 5500 LCMS/MS system. This system is equipped with a C18 reversed phase UHPLC column operated in negative mode using electrospray ionization [41]. The raw data was analyzed by AB SCIEX software (Analyst 1.6.2) with reduction of the data done in Microsoft Excel 2013. Analysis was done in a 96-well plate with two calibration curves and 6–8 quality control samples per batch. Samples were labeled BLOQ is they fell below the quantitation limit with ALOQ being labeled for samples above the quantitation limit.

**Sample preparation.** Each sample was accessioned into a LIMS system, assigned a unique identifier, and stored at -70˚C. To remove protein, dissociate small molecules bound to protein or trapped in the precipitated protein matrix, and to recover chemically diverse metabolites, proteins were precipitated with methanol, with vigorous shaking for 2 minutes (Glen Mills Genogrinder 2000) as described previously [42]. The resulting extract was divided into five fractions: two for analysis by ultra-high-performance liquid chromatography-tandem mass spectrometry run in positive mode (UPLC-MS/MS+; early and late; C18 column), two for

analysis by UPLC-MS/MS run in negative mode (UPLC-MS/MS-; C18 column and HILIC column), and one aliquot was retained for backup analysis, if needed.

**Mass spectrometry analysis.** Non-targeted UPLC-MS/MS were performed at Metabolon, Inc. as described [42]. The UPLC/MS/MS portion of the platform incorporates a Waters Acquity UPLC system and a Thermo-Finnegan LTQ mass spectrometer, including an electrospray ionization source and linear ion-trap [43] mass analyzer. Aliquots of the vacuum-dried sample were reconstituted, one each in acidic or basic LC-compatible solvents containing 8 or more injection standards at fixed concentrations (to both ensure injection and chromatographic consistency). Extracts were loaded onto columns (Waters UPLC BEH C18-2.1 x 100 mm, 1.7 μm) and gradient-eluted with water and 95% methanol containing 0.1% formic acid (acidic extracts) or 6.5 mM ammonium bicarbonate (basic extracts). The instrument was set to scan 99–1000 m/z and alternated between MS and MS/MS scans.

**Quality control.** All columns and reagents were purchased in bulk from a single lot to complete all related experiments. For monitoring of data quality and process variation, multiple replicates of extracts from a pool of human plasma were prepared in parallel and injected throughout the run, interspersed among the experimental samples. Instrument variability was determined by calculating the median relative standard deviation [27] for the standards that were added to each sample prior to injection into the mass spectrometers (median RSD = 4%; n = 21 standards). Overall process variability was determined by calculating the median RSD for all endogenous metabolites (i.e., non-instrument standards) present in 100% of technical replicate samples created from a homogeneous pool containing a small amount of all study samples (median RSD = 6%; n = 170 metabolites). In addition, process blanks and other quality control samples are spaced evenly among the injections for each day, and all experimental samples are randomly distributed throughout each day's run.

**Compound identification, quantification, and data curation.** Metabolites were identified by automated comparison of the ion features in the experimental samples to a reference library of chemical standard entries that included retention time, molecular weight (m/z), preferred adducts, and in-source fragments as well as associated MS spectra and curated by visual inspection for quality control using software developed at Metabolon [44]. Identification of known chemical entities was based on comparison to metabolomic library entries of more than 2,800 commercially-available purified standards. Subsequent QC and curation processes were utilized to ensure accurate, consistent identification and to minimize system artifacts, mis-assignments, and background noise. Library matches for each compound were verified for each sample. Peaks were quantified using area under the curve. Raw area counts for each metabolite in each sample were normalized to correct for variation resulting from instrument inter-day tuning differences by the median value for each run-day, therefore setting the medians to 1.0 for each run. This preserved variation between samples but allowed metabolites of widely different raw peak areas to be compared on a similar graphical scale. Missing values were imputed with the observed minimum after normalization.

**Bioinformatics.** The LIMS system encompasses sample accessioning, preparation, instrument analysis and reporting, and advanced data analysis. Additional informatics components include data extraction into a relational database and peak-identification software; proprietary data processing tools for QC and compound identification; and a collection of interpretation and visualization tools for use by data analysts. The hardware and software systems are built on a web-service platform utilizing Microsoft's.NET technologies which run on high-performance application servers and fiber-channel storage arrays in clusters to provide active failover and load-balancing.

## NanoString array

RNA was isolated from the hippocampus using RNeasy Plus kit following manufacturer's suggested protocol (Qiagen #74136). Quality and concentration of eluted RNA was measured by Nanodrop spectrophotometer. 200ng of total RNA per sample was quantified using a Nano-String array that consisted of 561 gene targets (Mouse Immunology v2 CodeSet). Following quality control, assay background subtraction (geometric mean of negative control values) and housekeeping normalization, there were a total of 318 genes that were examined for all samples. Raw RNA counts were analyzed for differential expression using NanoString's analysis software (nSolver 3.0). For the purpose of this experiment we considered only genes that showed at least a 2-fold change biologically relevant, following FDR-adjusted (Bejamini-Hochberg) multiple comparisons correction.

## Amyloid-β staining

Mouse brains were collected upon sacrifice and immediately put into a 10% Neutral Buffered Formalin for 24–48 hours. After this time period, the brains were transferred into 90% ethanol. Next, the brains were sent to the COCVD Pathology Research Core at the University of Kentucky to be embedded and sectioned onto microscope slides for immunohistochemistry. The sectioned tissue undergoes rehydration followed by tissue pretreatment in 90% formic acid for 3 minutes. The tissue was then treated with 3% $H_2O_2$ and 10% methanol for 30 minutes. Next, a M.O.M. Kit (Vector Laboratories, Inc. Burlingame, CA) was used following the standard protocol. Aβ was identified using an anti-Aβ$_{1-17}$ mouse monoclonal 6E10 antibody (1:3000; Signet Laboratories, Dedham, MA). Following this portion of the protocol, a DAB substrate kit (also Vector Laboratories, Burlingame, CA) was used for visualization. Next, a background stain utilizing NISSL was completed followed by dehydration. The slides were next imaged on the Aperio ScanScope XT Digital Slide Scanner System in the University of Kentucky Alzheimer's Disease Center Neuropathology Core Laboratory (20x magnificantion) and uploaded to the online database. Aperio ImageScope (version 12.3.2.8013) was used to analyze total anti-Aβ counted at 20x magnification (0.495px/um). 10 boxes (ROIs that are 600x600x600 microns) were randomly placed in each sample image and counted for percent positive Aβ (number of positive + number of strong positive/total number).

## Statistical analysis

All statistical analyses were completed using GraphPad Prism (GraphPad, San Diego, CA, USA). Two-tailed Student's *t*-test and 1-way ANOVA were performed for determination of differences between groups followed by Tukey's multiple comparisons test. Levels of statistical significance were reached when $p < 0.05$. For Metabolon, missing values in the data are assumed to be below the level of detection of the used instruments. Log transformations and imputation of missing values with the minimum observed values for each metabolite was conducted. This was followed by the usage of ANOVA to identify biochemicals that were significantly different between groups. Given the multiple comparisons inherent in analysis of metabolites, between-group relative differences are assessed using both p-value and false discovery rate analysis (q-value).

# Results

## Food intake, body weight, cognition and Aβ retention in *E4*FAD mice

After 16 weeks of feeding (when the mice reached 7 months of age), we found that *E4*FAD-inulin mice had significantly higher food intake than *E4*FAD-Control mice (Fig 1A), but did

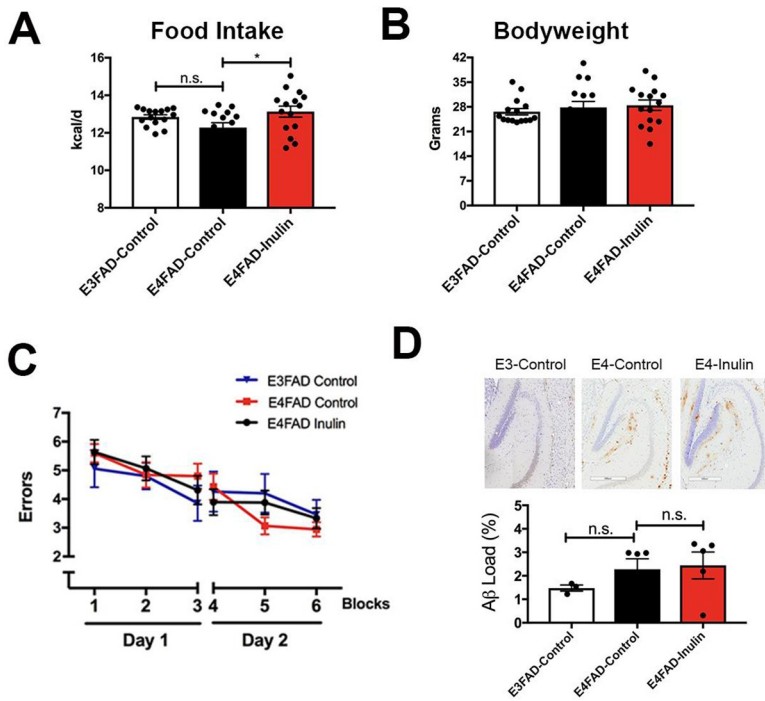

**Fig 1. Effect of inulin on food intake, body weight, cognition and Aβ retention in *E4*FAD mice.** Inulin intake increased daily energy uptake (kcal/d) from the diet in the *E4*FAD mice (A) but did not change bodyweight (B). (C) The mice showed no performance differences on the RAWM test. Wrong entries were recorded as errors. There were no significant differences in performance among *E3*FAD-Control, *E4*FAD-Control and *E4*FAD-Inulin mice in any of the 6 blocks. (D) Representative images of Aβ immunohistochemical staining from the three groups of mice (Top) and quantitation of the Aβ load (Bottom). There were no significant differences in Aβ load among the three groups of mice. Data are presented as the mean ± SEM. n.s. = not significant; $^*p < 0.05$.

not show any difference in bodyweight compared to *E4*FAD and *E3*FAD controls (Fig 1B). We assessed learning ability, spatial working and reference memory using RAWM behavioral tests, and did not find significant differences among the three groups during either the learning phase (Day 1, blocks 1–3) or the memory recall phase (Day 2, blocks 4–6; Fig 1C). Similarly, we did not find a significant difference in Aβ plaque aggregation among the three groups (Fig 1D). Our findings are consistent with published reports that *E4*FAD mice can be asymptomatic for AD up to 7 months of age [26].

## Inulin alters gut microbiome diversity in *E4*FAD mice

Interestingly, we found that the prebiotic inulin diet significantly altered gut microbiome diversity in *E4*FAD mice. Based on evaluation of the alpha (α) diversity metric (*i.e.*, Shannon index, H) of the fecal microbial community, *E4*FAD-Inulin mice had significantly lower diversity (at the genus level) than *E4*FAD-Control mice (Gaussian link function, p = 0.019; Fig 2A). By employing the Bray-Curtis dissimilarity metric followed by visualization of the data with an ordination plot, we observed that the fecal microbial community structures of *E4*FAD-Inulin mice were significantly different from that of the *E4*FAD-Control mice (ANOSIM R statistic = 0.877; p = 0.001; Fig 2B). Significant differences in fecal microbial community structure were also observed between *E3*FAD-Control and *E4*FAD-Control mice (ANOSIM R statistic = 0.017; p = 0.013, Fig 2C), but no significant difference was observed between the fecal microbial communities of FAD(-) and FAD(+) mice (p = ANOSIM R statistic = -0.011;

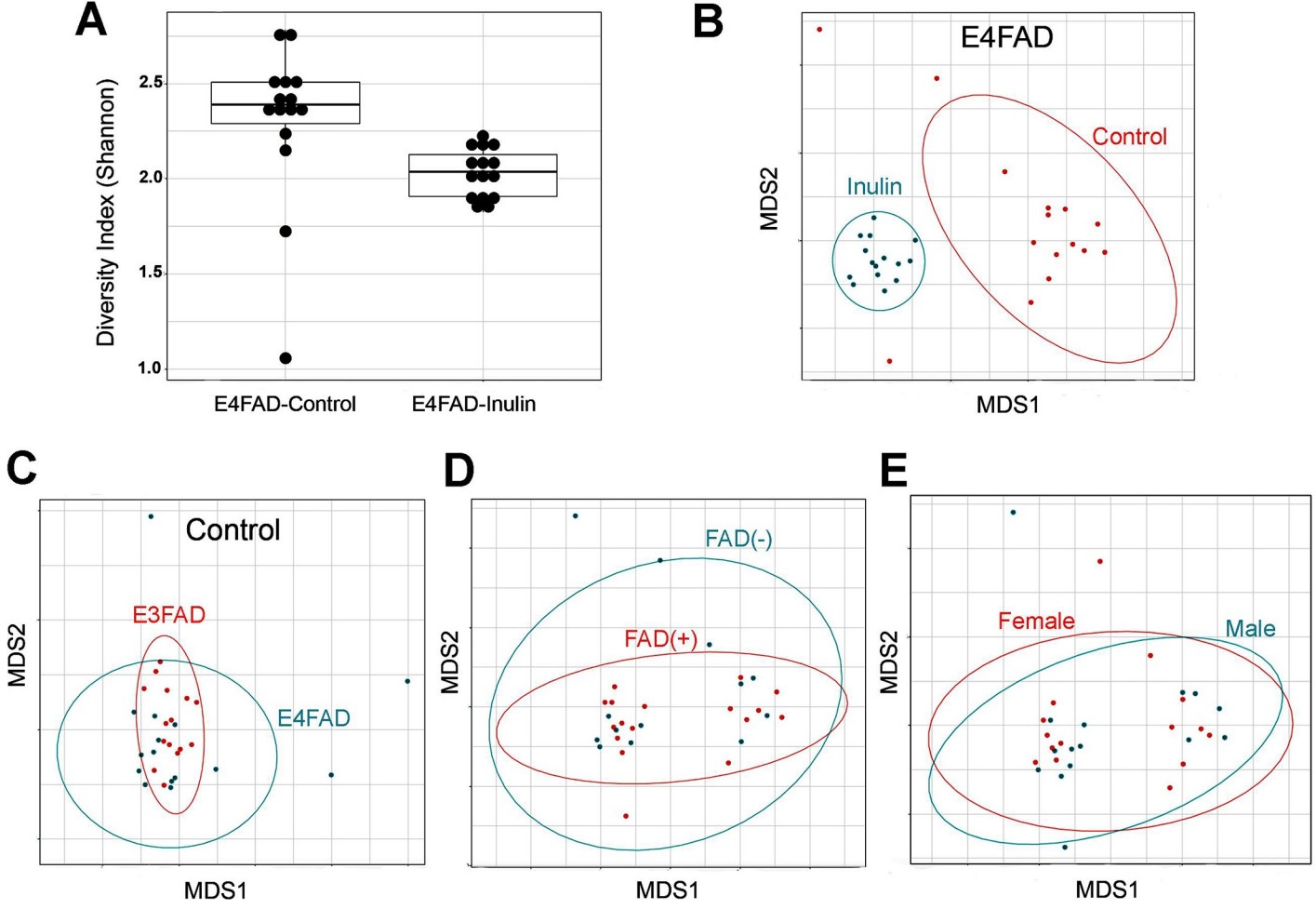

**Fig 2. Analysis of inulin-induced changes in gut microbiome diversity.** All analyses were performed on annotated sequence data at the genus level, rarefied to 10,000 sequences per sample. (A) *E4*FAD-Inulin mice had significantly lower α-diversity (shown here as the Shannon index) than *E4*FAD-Control mice (*p* = 0.019). (B) *E4*FAD-Control and *E4*FAD-Inulin mice exhibited a significant difference in β-diversity (p = 0.001). (C) *E3*FAD-Control and *E4*FAD-Control mice exhibited a significant difference in β-diversity (p = 0.013). (D) Mice with and without FAD mutations (FAD (+) vs. FAD(-)) exhibited no significant difference in β-diversity (p = 0.546). (E) Male and female mice did not exhibit a significant difference in β-diversity (p = 0.07).

p = 0.546, Fig 2D) or between *E4*FAD males and females across the two diets (ANOSIM R statistic = 0.066; p = 0.07, Fig 2E).

We further determined the fecal microbial taxa that were differentially abundant in inulin-fed versus control *E4*FAD mice (Table 2). Compared to *E4*FAD-Control mice, the relative abundance of bacteria from the genera *Prevotella* and *Lactobacillus* was elevated in *E4*FAD-Inulin mice, while the relative abundance of *Escherichia*, *Turicibacter* and *Akkermansia* was reduced in *E4*FAD-Inulin mice. *Prevotella spp.* are known to be involved in SCFA production and bile acid metabolism, and their relative abundance has been associated with improved glucose metabolism [45]. *Lactobacillus spp.* are commonly used as a probiotic [46]. Conversely, *Escherichia* and *Turicibacter spp.* have been associated with gastrointestinal (GI) inflammation, metabolic syndrome and diet-induced obesity [47]. Decreased abundance of *Akkermansia spp.* has recently been correlated with high dietary fiber intake and slower degradation of the colonic mucus barrier, thereby protecting against pathogens [48]. Thus, our results reveal that, overall, inulin increased the relative abundance of putatively beneficial gut microbiota and

**Table 2. Bacterial taxonomic analysis of inulin-induced changes in the gut microbiota of *E4*FAD mice.**

| OTU | *E4* Inulin vs. *E4* control | *E4* control vs. *E3* control | Potential Function/Association | Citations |
|---|---|---|---|---|
| | FDRC (Fold Change) | FDRC (Fold Change) | | |
| k__Bacteria; p__Bacteroidetes; c__Bacteroidia; o__Bacteroidales; f__Paraprevotellaceae; g__*Prevotella* | 2.17E-03(+1.78) | _ | Involved in bile acid metabolism; Has a negative correlation with BMI; Involved in SCFA production; Increased by a high fiber diet; Exerts a protective effect against diabetes | [49] |
| k__Bacteria; p__Firmicutes; c__Bacilli; o__Lactobacillales; f__Lactobacillaceae; g__*Lactobacillus* | 0.02 (+1.48) | _ | Common probiotic; Inhibits pathogens; produces lactate | [46] |
| k__Bacteria; p__Proteobacteria; c__Gammaproteobacteria; o__Enterobacteriales; f__Enterobacteriaceae; g__*Escherichia* | 8.76E-04 (-3.28) | _ | Involved in GI disorders and inflammation | [47] |
| k__Bacteria; p__Firmicutes; c__Bacilli; o__Turicibacterales; f__Turicibacteraceae; g__*Turicibacter* | 0.03 (-0.47) | _ | Implicated to have a role in metabolic syndrome; may play a role in abnormal metabolism in Type 2 Diabetes | [50] |
| k__Bacteria; p__Proteobacteria; c__Gammaproteobacteria; o__Enterobacteriales; f__Enterobacteriaceae; g__*Proteus* | 0.02 (-4.54) | 0.01 (7.64) | Strong association with Chron's disease; Possess many virulence factors relevant to GI disease (motility, adherence, urase production, IgA proteases, antibiotic resistance); Has also been linked to gastroenteritis | [51] |
| k__Bacteria; p__Verrucomicrobia; c__Verrucomicrobiae; o__Verrucomicrobiales; f__Verrucomicrobiaceae; g__*Akkermansia* | 1.82E-04 (-0.97) | - | Mucin degraders; shown to decrease on a high-fiber diet, thereby protecting the mucus layer and intestinal barrier | [48] |

The FDRC is the Benjamini-Hochberg false-discovery rate (FDR)-adjusted *p*-value. Only those genus-level taxa with an FDRC <0.05 are shown.

reduced that of putatively proinflammatory taxa within the context of reduced overall α-diversity.

### Inulin enhances systemic metabolism in *E4*FAD mice

**Gut (cecum).** SCFA measurement in the cecum by metabolomic analysis revealed significant increases in cecal acetate (Fig 3A), butyrate (Fig 3B), and propionate (Fig 3C) in

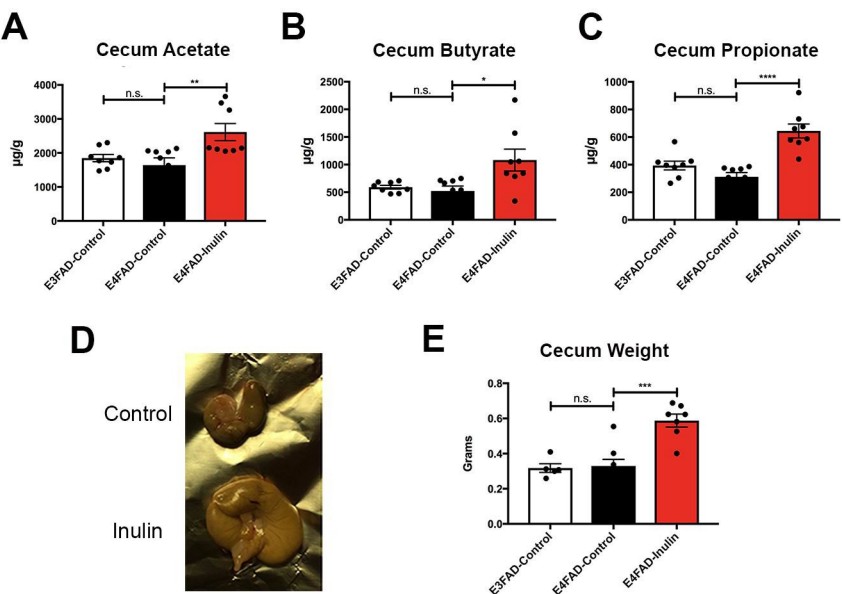

**Fig 3. Analysis of inulin-induced changes in metabolism and weight in the cecum.** *E4*FAD-Inulin fed mice had significantly enhanced production of the short-chain fatty acids (A) acetate; (B) butyrate; and (C) propionate compared to the controls. *E3*FAD-Control and *E4*FAD-Control mice did not exhibit significant differences in these measures. (D) Representative images showing the significantly increased cecum size of the inulin-fed *E4*FAD mice compared to the *E4*FAD control mice. (E) Quantitation of data in (D) comparing the cecum weights among the three groups. Data are presented as the mean ± SEM. n.s. = not significant; $^*p < 0.05$; $^{**}p < 0.01$; $^{***}p < 0.001$; $^{****}p < 0.0001$.

*E4*FAD-Inulin mice compared to *E4*FAD-Control mice. These results suggest that dietary supplementation with inulin causes increased fermentation in the cecum that leads to increased production of SCFAs. Additionally, when the cecum was excised at the end of the study, we observed that *E4*FAD-Inulin mice had significantly enlarged ceca compared to the controls (Fig 3D and 3E). This is likely due to the cecum being the primary site of inulin fermentation [52]. Enlargement of the cecum has been suggested to be a result of increased microbial content [53], which is consistent with our findings.

**Periphery (blood).** We next examined whether microbiota-related metabolites were released from the gut to the bloodstream. Since SCFAs, tryptophan metabolites, and bile acids are the three most studied categories of metabolites in host-microbiota interactions [54], we chose to measure these metabolites in mice peripheral blood. We found that acetate was significantly increased in blood in *E4*FAD-Inulin mice compared to *E4*FAD-control mice (Fig 4A). Although butyrate (Fig 4B) and propionate (Fig 4C) were also increased in the bloodstream of *E4*FAD-Inulin mice, the increase was not as substantial as in the cecum. Metabolites related to tryptophan metabolism, including indolepropionate (IPA) (Fig 4D) and indoleacrylate (Fig 4E), were observed at much higher levels in *E4*FAD-Inulin mice than in *E4*FAD-Control or *E3*FAD-Control mice. *E4*FAD-Inulin mice also had significantly elevated levels of bile acids, including cholate (Fig 4F) and deoxycholate (Fig 4G), compared to *E4*FAD-Control mice.

In addition to microbial metabolites, we found that inulin enhanced the levels of metabolites involved in the mitochondrial tricarboxylic acid (TCA) cycle (Fig 5A). Succinate (Fig 5B), fumarate (Fig 5C), and malate (Fig 5D) were significantly increased in *E4*FAD-Inulin mice compared to *E3*FAD-Control mice. In contrast, *E4*FAD-Control mice exhibited decreased levels of these metabolites in comparison to *E3*FAD-Control mice, consistent with published reports that *APOE4* carriers had early-onset mitochondrial dysfunction compared to

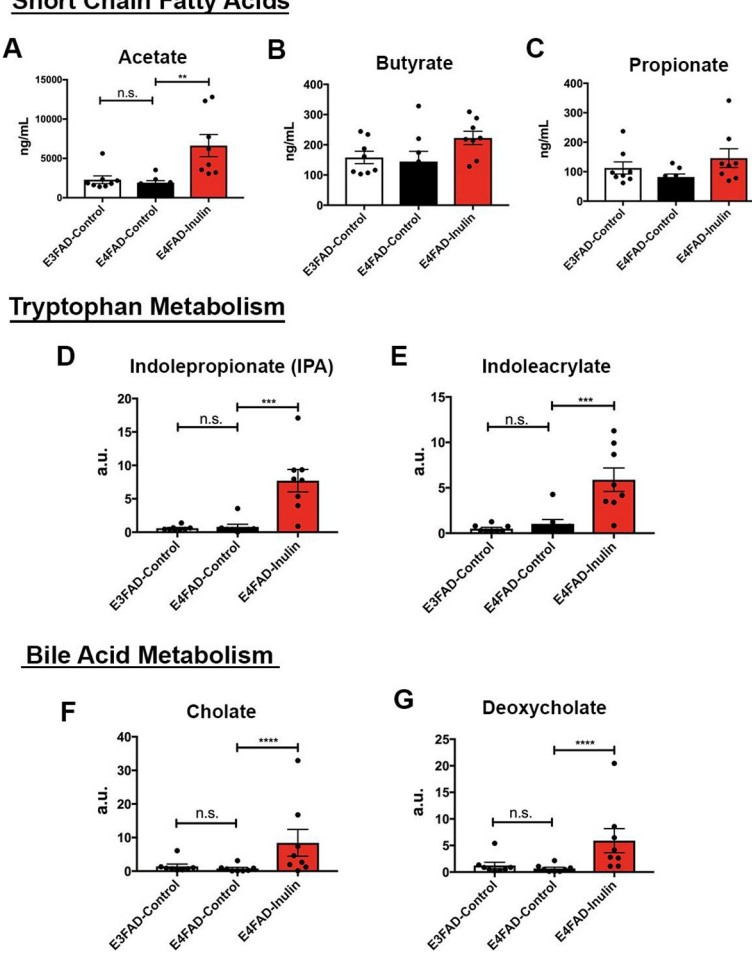

**Fig 4. Analysis of inulin-induced changes in microbial metabolism changes in the periphery.** *E4*FAD-Inulin mice had a significant increase in (A) acetate and trend-level increases in (B) butyrate and (C) propionate. *E4*FAD-Inulin mice had increased tryptophan metabolism, as indicated by the levels of (D) indolepropionate (IPA) and (E) indoleacrylate, and increased bile acid metabolism, as indicated by the levels of (F) cholate and (G) deoxycholate. n.s. = not significant; $^{**}p < 0.01$; $^{***}p < 0.001$; $^{****}p < 0.0001$.

noncarriers [8]. In addition to entering the TCA cycle, glucose in the brain can be metabolized via the PPP to generate NADPH—the major reducing equivalent in the cell that supports glutathione- and thioredoxin-dependent antioxidant systems [55] (Fig 5A). In line with this, we found that inulin also enhanced the levels of metabolites generated via the PPP, including ribose (Fig 5E), ribulose (Fig 5F), ribonate (Fig 5G), and arabonate (Fig 5H), suggesting that dietary supplementation with inulin may reduce cellular oxidative stress and enhance antioxidant function.

**Brain (hippocampus).** Using *in vivo* $^1$H-MRS to acquire spectra from the hippocampal voxel of experimental mice (Fig 6A), we found that the levels of scyllo-inositol, a microbiota-produced metabolite, was increased in the hippocampus of *E4*FAD-Inulin mice, compared to control mice (Fig 6B and 6C). The results indicate that the scyllo-inositol produced by the gut microbiome in response to dietary inulin can reach the hippocampus. We also observed that the increase in scyllo-inositol was accompanied by a decrease in myo-inositol (Fig 6D). Scyllo-inositol is a microbiota-produced metabolite and has been used to inhibit Aβ aggregation in

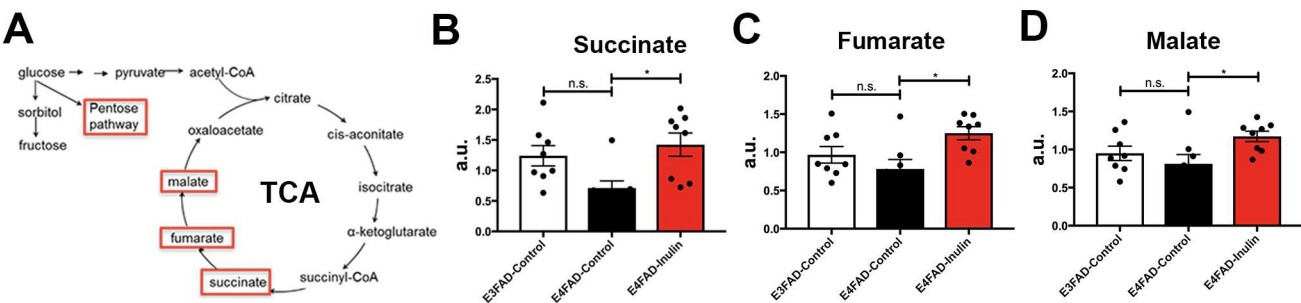

## Metabolites related to Pentose Phosphate Pathway (PPP)

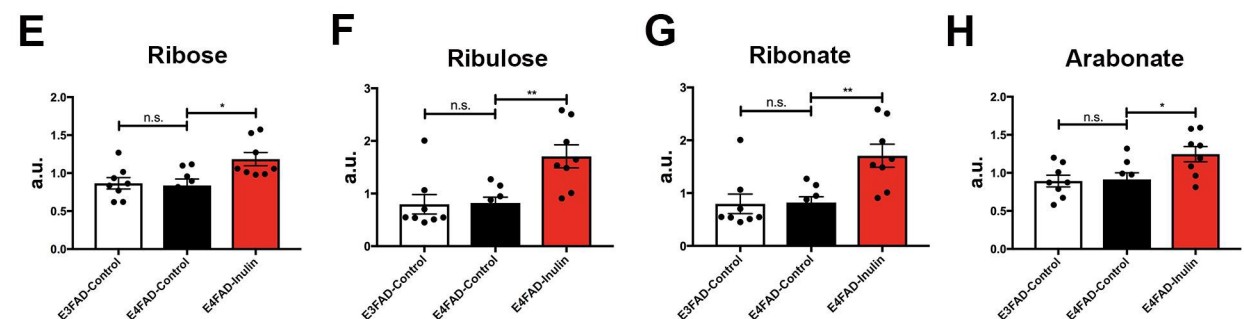

**Fig 5. Analysis of inulin-induced changes in glycolytic metabolism in the periphery.** (A) An illustration of the glycolytic pathway through the TCA cycle and PPP. *E4*FAD mice fed prebiotic inulin showed changes in the levels of TCA cycle metabolites, including (B) succinate, (C) fumarate, and (D) malate, compared to *E4*FAD mice fed the control diet. *E4*FAD mice fed the prebiotic inulin showed a significant increase in the levels PPP-associated metabolites including (E) ribose, (F) ribulose, (G) ribonate, and (H) arabonate compared to *E4*FAD mice fed the control diet. n.s. = not significant; $^*p < 0.05$; $^{**}p < 0.01$.

clinical trials [56], while myo-inositol has been shown to be increased in *APOE4* carriers with preclinical AD before detection of elevated Aβ levels [57]. Administration of scyllo-inositol has been shown to lead to a reduction in elevated brain myo-inositol levels in AD [58].

### Inulin decreases brain inflammation in *E4*FAD mice

Since Familial Alzheimer's disease (FAD) mutations result in increased production of Aβ leading to neuroinflammation [59], we decided to examine whether dietary inulin impacts inflammatory gene expression in the hippocampus of *APOE4* mice with FAD mutations. We found that among 318 genes, 52 were enriched in *APOE4* mice with FAD mutations (Fig 7A). This suggests that the FAD mutations do indeed significantly enhance neuroinflammation in the hippocampus of the *E4*FAD mice. Among the 52 enriched genes, the greatest change in expression was observed in 4 genes that had a 2-fold lower level of expression in *E4*FAD-Inulin mice compared to *E4*FAD-Control mice (Fig 7B). Two of these genes that were significantly decreased in expression were chemokine (C-C motif) ligand 4 (*CCL4*; MGI:98261) (Fig 7C) and Fc receptor IgG low affinity 4 (*Fcgr4*; MGI:2179523) (Fig 7D). C-X-C motif chemokine 10 (*CXCL10*; MGI:1352450) (Fig 7E) and integrin alpha X (*Itgax*; MGI:96609) (Fig 7F) had a trend toward significant. Notably, these four proinflammatory genes were minimally expressed in *APOE4* mice without FAD mutations, implying their importance in AD. Along with this, differences in diet did not show any significant effects (Fig 7C–7F).

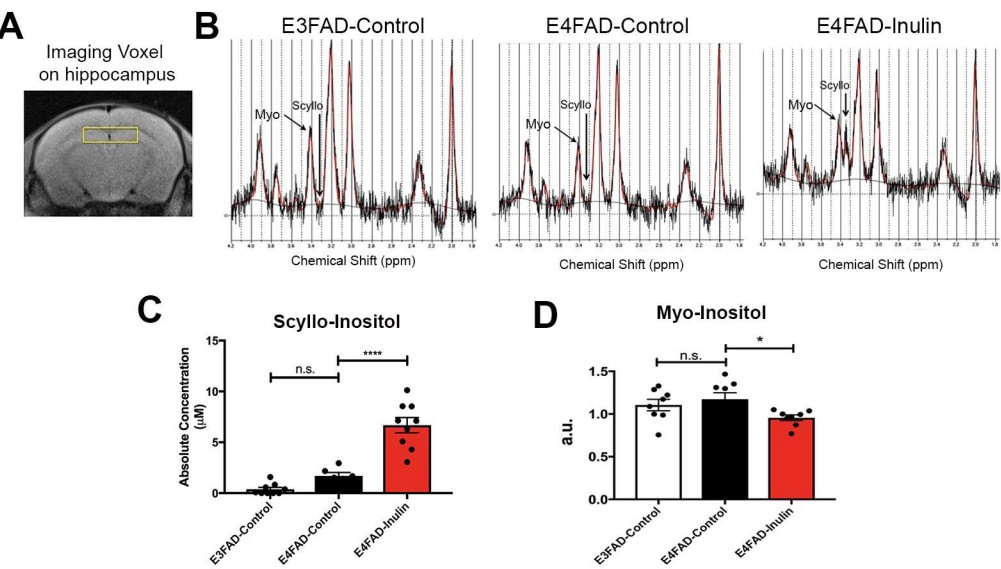

**Fig 6. Analysis of inulin-induced changes in microbial metabolism in the brain.** Scyllo-inositol was dramatically increased in the hippocampus of *E4*FAD mice fed the prebiotic inulin compared to the *E4*FAD mice fed the control diet. (A) Shows the voxel that was used for MRS. Representative spectra are shown for (B) the *E3*FAD-Control, *E4*FAD-Control, and *E4*FAD-Inulin groups. The arrow indicates scyllo-inositol, which was dramatically increased in mice fed the prebiotic inulin. Scyllo-inositol (C) was significantly increased in E4FAD inulin mice compared to control fed E3FAD and E4FAD mice. Myo-inositol (D) was significantly decreased in E4FAD mice fed inulin. Data are the mean ± SEM, n.s. = not significant; $^{****}p < 0.0001$.

## Discussion

*APOE4* carriers may develop systemic metabolic dysfunction decades before the onset of Alzheimer's disease. Early interventions to protect and preserve metabolic functions in asymptomatic *APOE4* carriers are critical to reduce or eliminate the risk of AD. In this study, we demonstrate that a diet supplemented with the prebiotic inulin leads to an altered gut microbiome, increased microbial metabolism via gut-brain axis components, enhanced glycolytic metabolism in the periphery, and reduced hippocampal inflammatory gene expression in asymptomatic *E4*FAD mice. The findings are consistent with literature, showing that targeting the gut microbiome with inulin can modulate the peripheral immune response and alter neuroinflammation in middle age mice [60], and reduce oxidative stress and prevent neurodegeneration as well as atherosclerosis in mice fed with high-fat diet [14, 15].

Our analyses of the gut microbiome revealed that mice fed an inulin-supplemented diet display increased abundance of *Prevotella* and *Lactobacillus spp*. Consistent with reports that *Prevotella spp.* are involved in the production of SCFAs, bile acids, and succinate [49], we observed an elevation of SCFAs in the cecum and the blood, and an elevation of bile acids and succinate in the blood of inulin- fed mice. SCFAs play a major role in modulating glucose homeostasis. In particular, butyrate has been shown to improve mitochondrial function, increase glucose sensitivity and decrease inflammation; acetate has been shown to decrease appetite and body fat and to increase mitochondrial efficiency; and propionate has been shown to increase glucose sensitivity, regulate body weight and modulate lipid metabolism [61]. Bile acids regulate lipid, glucose and energy homeostasis as well as their own synthesis, detoxification and transport [62]. Changes in SCFA and bile acid levels in inulin-fed *E4*AD mice suggest improved glucose metabolism and mitochondrial function. We found increased metabolites related to glycolytic metabolism, including components of the mitochondrial TCA

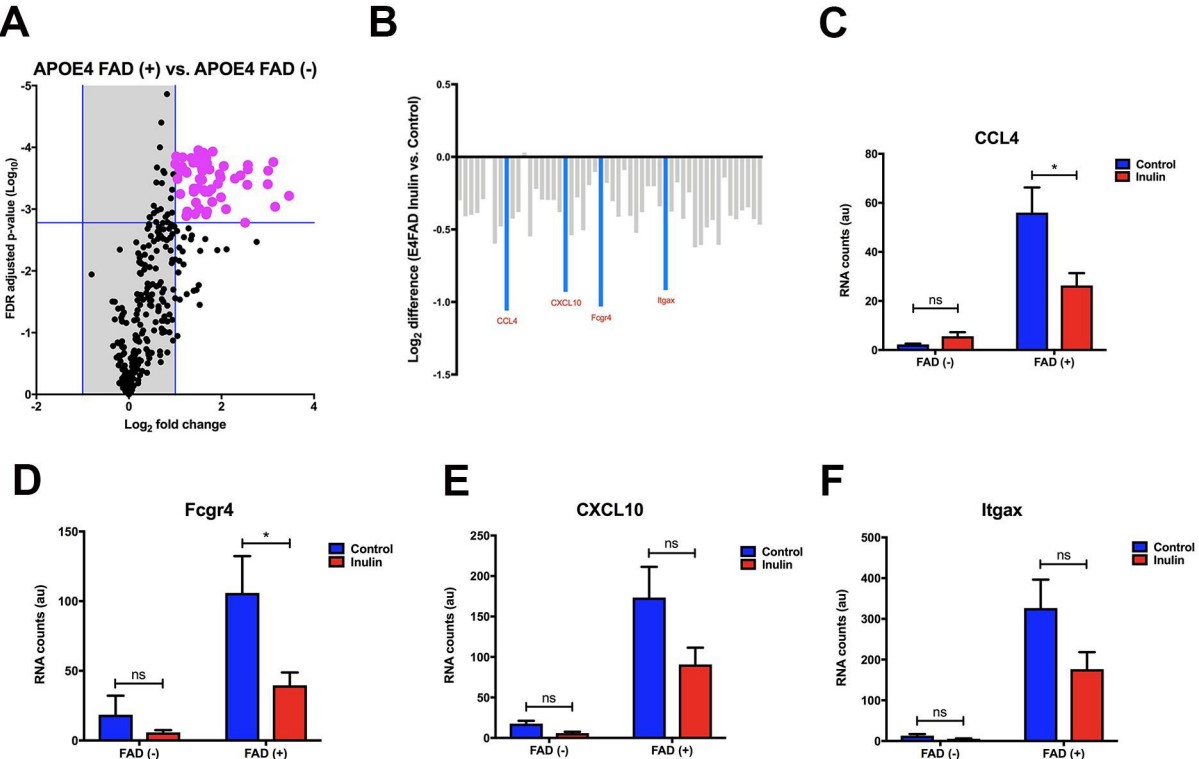

**Fig 7. Analysis of inulin-induced changes in inflammatory gene expression in the hippocampus.** (A) In *APOE4* mice with and without the FAD mutations, data are plotted as the mean expression ratios with their respective FDR-adjusted *p*-value. This revealed 52 genes that were significantly differentially enriched (magenta) with $\log_2$ (fold change) greater than one due to the FAD mutations. (B) The 52 genes that were enriched due to the FAD mutations were further examined in the context of diet manipulation. Overall, there was a trend for most of the 52 genes to be downregulated due to the prebiotic inulin in *E4*FAD mice (e.g., gray bars). Furthermore, an approximately 2-fold decrease in expression was observed in 4 genes (aquamarine). Among these, (C) *CCL4*, and (D) *Fcgr4* were significantly decreased. Although there was a strong visual trend for both (E) *CXCL10* and (F) *Itgax* to be decreased due to the prebiotic inulin, these decreases were not statistically significant. Data are mean ± SEM, n.s. = not significant; *$p < 0.05$.

cycle, such as succinate, as well as components of the PPP. As the PPP is a major antioxidant system, increases in *Prevotella* may indicate an enhancement of immune function. Similarly, increases in the relative abundance of *Lactobacillus* may also relate to improved immune function. *Lactobacillus spp.* are commonly used as probiotics to improve digestion and boost immunity, and can also help inhibit the growth of harmful bacteria and improve intestinal flora balance [63]. This is consistent with our findings that the relative abundance of several potentially harmful microbial taxa was decreased in *E4*FAD-Inulin mice relative to *E4*FAD--Control mice, including *Escherichia*, and *Turicibacter* and *Proteus*. While the relative abundance of *Proteus* was substantially higher in *E4*FAD-Control mice than in *E3*FAD-Control animals, this abundance was significantly reduced in *E4*FAD-Inulin mice. The relative abundance of *Akkermansia spp* that has been shown to have both positive and negative effects on host health, was also decreased in *E4*FAD-Inulin relative to *E4*FAD-Control mice. In a recent study of a low-fiber diet, *Akkermansia spp.* were shown to increase in abundance and express mucin-degrading enzymes [48]. Mucin degradation that leads to gut barrier degradation and increased susceptibility to pathogens, was mitigated by a high-fiber diet [48]. The reduced relative abundance of *Akkermansia* in *E4*FAD-Inulin mice relative to *E4*FAD-Control mice could indicate a protective role of fiber through both a decrease in the abundance and activity of mucin-degrading species and a decrease in the general production of SCFAs, leading to

increased gut barrier integrity. Collectively, our findings of inulin-induced alterations in the gut microbiome demonstrate potential enhancements in systemic metabolic and immune functions.

We show that α-diversity is lower in the *E4*FAD-Inulin mice than in the *E4*FAD-Control mice. Although an increase in α-diversity has been reported to be beneficial in obesity and type 2 diabetes mellitus (T2DM) [64], our group and others have found the opposite to be true in mouse models for AD [65–67]. We reported that aging-induced dysbiosis (gut microbiome imbalance) increases α-diversity in aged mice [65] while mice fed a ketogenic diet had reduced α-diversity with increased beneficial microbiota [66]. Similarly, our current study indicates that dietary supplementation with inulin increases the relative abundance of putatively beneficial gut microbiota and reduces that of putatively proinflammatory taxa in the context of reduced overall α-diversity.

We report a significant increase in scyllo-inositol and decrease in myo-inositol in the hippocampus of the *E4*FAD-Inulin mice. Scyllo-inositol has been reported to be produced by *Bacillus subtilis* [68] and has been used to inhibit Aβ aggregation in clinical trials [56]. In contrast, myo-inositol is considered an inflammatory marker, and a dramatic increase in myo-inositol suggests increased demyelination and proliferation of glial cells in inflammation [69]. Our findings indicate that dietary inulin can improve immune function and reduce Aβ aggregation. This is consistent with reduced proinflammatory gene expression in the hippocampus of *E4*FAD-Inulin mice with FAD mutations. We report significant decreases in *CCL4* and *Fcgr4* and a decreased trend for *CXCL10* and *Itgax*. *CCL4* is a chemokine that has been found to be altered in human *APOE4* carriers and is associated with the accumulation of Aβ in the brain [70]. *Fcgr4*, along with other member of the *FCGR* family, is upregulated under inflammatory conditions and potentially stimulates vascular damage and neurodegeneration [71]. *CXCL10* is a chemokine found in high concentrations in AD, and cerebrospinal fluid *CXCL10* concentrations have been positively correlated with cognitive impairment [72]. Finally, *Itgax* is an inflammatory integrin protein that appears to be induced in degenerative 'MGnD' microglia in AD [43]. These gene expression data are consistent with *in vivo* brain metabolism data, which show increased scyllo-inositol (Aβ inhibition) and reduced myo-inositol (decreased inflammation). Taken together, our results show that an inulin-enriched diet improves immune function not only in the periphery but also in the CNS. Our findings suggest that early intervention with a prebiotic diet systemically enhances metabolism and reduces inflammation.

The systemic changes produced by dietary inulin may help reduce the risk for AD over time. Although AD is a neurodegenerative disorder, accumulating evidence suggests that systemic metabolic dysfunction and inflammation contribute greatly to AD risk. This leaves individuals with insulin resistance, T2DM, hyperlipidemia, obesity, or other metabolic diseases at an increased risk for AD [3]. SCFA deficiency has been associated with T2DM; metabolic dysfunction including inflammation, insulin resistance and endoplasmic reticulum stress, are known to underlie T2DM in peripheral tissues [73]. A similar condition has also been described in the brains of AD patients. In line with this, some groups have even described AD as Type 3 diabetes mellitus (T3DM) [74]. Interventions that could alleviate inflammation and restore metabolic function systemically may ultimately reduce the risk of T2DM and AD. Interestingly, inulin was recently shown to have beneficial effects towards metabolic syndrome in individuals with T2DM by reducing the levels of proinflammatory cytokines (e.g., TNF-α and IL-6) and insulin resistance, thereby lowering blood pressure and improving the lipid profile and glucose homeostasis [75]. Other clinical trials have shown similarly that increased intake of dietary fiber (indigestible but fermentable carbohydrates) was sufficient to induce metabolic improvements in patients with T2DM [8–10, 76]. In line with this, a recent study

showed that probiotics were able to counteract AD progression by modulating gut microbiome, reducing inflammatory cytokines, increasing SCFAs and elevating neuroprotective gut hormones [16]. These findings in humans are consistent with the results from our current mouse study. Recent studies demonstrating that treatment of systemic inflammation and metabolic dysfunction can reverse cognitive decline and prevent development of AD [77] encourage us to anticipate that long-term inulin treatment can reduce the risk of developing AD.

In the present study, we show that inulin was able to enhance systemic metabolism by modulating gut microbiome even before the development of Aβ in the E4FAD mice. In the future, it will be interesting to know if inulin is also able to halt the progression of Aβ after AD has developed. It will also be interesting to identify if inulin is effective in counteracting other AD pathologies, such as tau tangles, with other animal models. Finally, it will important to determine whether APOE4 protein level also changes due to Inulin and whether the metabolic responses to inulin are dose-dependent (e.g., 4% vs 8% inulin).

## Conclusion

In this study, we detail how a dietary intervention with inulin, a prebiotic, can effectively alter the gut microbiome in a preclinical AD mouse model, enhance systemic metabolic functions and reduce brain inflammation characteristic of early AD. We focus on the underlying metabolic features contributed by the *APOE ε4* genotype, the largest risk factor for the development of AD, and show how dietary inulin intervention can mitigate them. This represents a shift in AD research, from the amyloid hypothesis to the metabolic features underlying the disease. As impairment of energy metabolism and elevation of neuroinflammation are well-established features of AD, a prebiotic-rich diet might be a potentially useful approach to prevent the onset of the disease. For future studies, it will be important to establish whether an inulin-enriched diet can inhibit Aβ aggregation and impede cognitive decline in symptomatic *E4*FAD mice. Understanding the dietary effects in the context of the gut-brain axis may have significant future implications for preventing AD in asymptomatic *APOE4* carriers.

## Author Contributions

**Conceptualization:** Jared D. Hoffman, Tyler C. Hammond, Stefan J. Green, Ai-Ling Lin.

**Data curation:** Jared D. Hoffman, Lucille M. Yanckello, George Chlipala.

**Formal analysis:** Jared D. Hoffman, Lucille M. Yanckello, George Chlipala, Scott D. McCulloch, Sydney Sun, Josh M. Morganti.

**Funding acquisition:** Ai-Ling Lin.

**Investigation:** Jared D. Hoffman, George Chlipala, Scott D. McCulloch, Ishita Parikh, Josh M. Morganti, Stefan J. Green, Ai-Ling Lin.

**Methodology:** Jared D. Hoffman, Lucille M. Yanckello, George Chlipala, Scott D. McCulloch, Ishita Parikh, Josh M. Morganti, Stefan J. Green, Ai-Ling Lin.

**Project administration:** Stefan J. Green, Ai-Ling Lin.

**Resources:** George Chlipala, Scott D. McCulloch, Josh M. Morganti, Ai-Ling Lin.

**Software:** George Chlipala, Scott D. McCulloch, Ai-Ling Lin.

**Supervision:** Ishita Parikh, Stefan J. Green, Ai-Ling Lin.

**Validation:** Lucille M. Yanckello, Tyler C. Hammond, Sydney Sun, Ai-Ling Lin.

**Visualization:** Lucille M. Yanckello, Scott D. McCulloch, Sydney Sun, Ai-Ling Lin.

**Writing – original draft:** Jared D. Hoffman, George Chlipala, Ai-Ling Lin.

**Writing – review & editing:** Jared D. Hoffman, Lucille M. Yanckello, George Chlipala, Tyler C. Hammond, Scott D. McCulloch, Ishita Parikh, Sydney Sun, Josh M. Morganti, Stefan J. Green, Ai-Ling Lin.

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
