## [Decision Letter · Decision Letter 0]

30 Jul 2019

PONE-D-19-17293

Dietary inulin alters the gut microbiome, enhances systemic metabolism and reduces neuroinflammation in an APOE4 relevant mouse model of preclinical Alzheimer’s Disease

PLOS ONE

Dear Prof. Ai-Ling Lin,

Thank you for submitting your manuscript to PLOS ONE. After careful consideration, we feel that it has merit but does not fully meet PLOS ONE’s publication criteria as it currently stands. Therefore, we invite you to submit a revised version of the manuscript that addresses the points raised during the review process.

We would appreciate receiving your revised manuscript by 13th of September. To enhance the reproducibility of your results, we recommend that if applicable you deposit your laboratory protocols in protocols.io, where a protocol can be assigned its own identifier (DOI) such that it can be cited independently in the future. For instructions see: http://journals.plos.org/plosone/s/submission-guidelines#loc-laboratory-protocols

We look forward to receiving your revised manuscript.

Kind regards,

Florian Reichmann, M.D., Ph.D.

Academic Editor

PLOS ONE

Journal Requirements:

2. We note that your article has been submitted as a "Collection Review" article type, but is primary research submitted for a Call for Papers. When resubmitting your manuscript, we ask that you update your article type to "Research Article" in the online submission form. Please note that some fields in the submission form, particularly in the "Additional Information" field, will have been reset with this change, so please go through your submission in full to ensure that all information is accurate and complete when resubmitting your manuscript.

3. At this time, we request that you please include the method of euthanasia and the source of the animals in your Methods section. We also ask that you please include your ethics information " All experimental procedures were performed according to NIH guidelines and approved by the Institutional Animal Care and Use Committee (IACUC) at the University of Kentucky (UK)." in the on-line ethics statement on the details page. Thank you for your attention to these requests.

[Scott McCulloch is employed by Metabolon Inc. All other authors have declared that no competing interests exist.].   

We note that one or more of the authors are employed by a commercial company: 'Metabolon Inc'.

Additional Editor Comments (if provided):

Reviewer #1: This is a very interesting paper showing how preserving metabolic function through microbiota modulation AD risk can be reduced.

In my opinion the author should mention recent papers demonstrating that upon probiotics administration AD progression is counteracted based on microbiota modulation, reduction of inflammatory cytokinesm increased SCFAs, increased neuroprotective gut hormones (Bonfili L. et al. Scientific Reports 2017)

Reviewer #2: 1 The expression of APOE4 should be included after the adminstration of inulin. And more AD biomaker as Tau, BACE, p-Tau, or histopathology of brain and colon.

2 I think there are at lest two does of inulin treated groups should be proved.

3 What the dose judgment, 8% of inulin? AS: Vishal Singh et al., (2018), Dysregulated Microbial Fermentation of Soluble Fiber Induces Cholestatic Liver Cancer, Cell, DOI: https://doi.org/10.1016/j.cell.2018.09.004.

4 More the correction analysis bteween the AD biomaker and gut microbiota, Metabolite.

5 Are the metabolite of Scyllo-inositol and myo-inositol is related to AD?

Reviewer #3: In their manuscript, the Authors have investigated the neuroprotective role of inulin ingestion by in vivo tests on E4FAD mice. The Authors have demonstrated that inulin has beneficial effects on microbiota and neuroinflammation. In fact, the experiments were performed to demonstrate that inulin compound have an anti-neurodegenerative role in mice fed with inulin diet. The topic is really interesting and the experiments are well designed. For this reason, I suggest only some corrections.

1. The references Nuzzo D, et al. Nutrients 2018 and Amato A et al. Nutrients 2017 should be added in the introduction to describe the link between oxidative stress and diseases.

2. I suggest to give more information regarding the mouse treatment. The Authors may add some additional information like food consumption (g for a day) and total caloric intake. How many grams of inulin are daily supplied to mice?

3. In Figure 1(D) the image of E3-control is missing.

4. The title of the article is inappropriate because the data reported are insufficient to claim a potential role of inulin for the Alzheimer disease. I suggest to remove "[…] relevant mouse model of preclinical Alzheimer's Disease" from the title.
---

## [Author Response · Author response to Decision Letter 0]

12 Aug 2019

We have included the point-by-point response to the reviewer's comments in a separate file. We also addressed to the editor's request by adding the method of euthanasia and the source of the animals in our Methods section.

---

## [Editor Report · Decision Letter 1]

16 Aug 2019

Dietary inulin alters the gut microbiome, enhances systemic metabolism and reduces neuroinflammation in an APOE4 mouse model

PONE-D-19-17293R1

Dear Dr. Ai-Ling Lin,

We are pleased to inform you that your manuscript has been judged scientifically suitable for publication and will be formally accepted for publication once it complies with all outstanding technical requirements.

With kind regards,

Florian Reichmann, M.D., Ph.D.

Academic Editor

PLOS ONE
---

## [Editor Report · Acceptance letter]

21 Aug 2019

PONE-D-19-17293R1 

Dietary inulin alters the gut microbiome, enhances systemic metabolism and reduces neuroinflammation in an *APOE4* mouse model 

Dear Dr. Lin:

I am pleased to inform you that your manuscript has been deemed suitable for publication in PLOS ONE. Congratulations! Your manuscript is now with our production department. 

With kind regards,

on behalf of

Dr. Florian Reichmann 

Academic Editor

PLOS ONE